# Examining Coping Strategies and Their Relation with Anxiety: Implications for Children Diagnosed with Cancer or Type 1 Diabetes and Their Caregivers

**DOI:** 10.3390/ijerph21010077

**Published:** 2024-01-10

**Authors:** Silvia Spaggiari, Giulia Calignano, Maria Montanaro, Silvana Zaffani, Valerio Cecinati, Claudio Maffeis, Daniela Di Riso

**Affiliations:** 1Department of Developmental Psychology and Socialization (DPSS), University of Padua, 35131 Padova, Italy; giulia.calignano@unipd.it (G.C.); daniela.diriso@unipd.it (D.D.R.); 2Complex Structure of Pediatrics and Pediatric Oncohematology “Nadia Toffa”, Central Hospital Santissima Annunziata, 74121 Taranto, Italy; maria.montanaro@libero.it (M.M.); valerio.cecinati@asl.taranto.it (V.C.); 3Pediatric Diabetes and Metabolic Disorders, Department of Surgical Sciences, Dentistry, Paediatrics and Gynaecology, University of Verona, 37134 Verona, Italy; silvana.zaffani@aovr.veneto.it (S.Z.); claudio.maffeis@univr.it (C.M.)

**Keywords:** pediatric type 1 diabetes (T1D), pediatric cancer, coping strategies, children’s anxiety, caregivers’ anxiety

## Abstract

The onset of chronic diseases in childhood represents a stressful event for both young patients and their caregivers. In this context, coping strategies play a fundamental role in dealing with illness-related challenges. Although numerous studies have explored coping strategies employed by parents of children with chronic diseases, there remains a gap in the understanding of children’s coping strategies and their correlation with their and their parents’ anxiety. This study aims to investigate coping strategies and their interaction with anxiety in groups of young patients with cancer, type 1 diabetes (T1D), and their respective caregivers, in comparison to healthy children and caregivers. We recruited a total of 61 control children, 33 with cancer, and 56 with T1D, 7 to 15 years old, along with their mothers. Each participant completed a customized survey and standardized questionnaires. No significant differences emerged in coping strategies used by children among the different groups. However, when examining the association between coping strategy and anxiety, we found specific patterns of interaction between children’s use of coping strategies and their and their mothers’ anxiety levels. This study underscores the importance of an illness-specific approach to gain deeper insights into this topic and develop targeted interventions aimed at enhancing the psychological well-being of these vulnerable populations.

## 1. Introduction

Having a chronic disease in childhood and adolescence is a stressful condition for both young patients and their parents. Physical pain, dealing with symptoms, medical treatment, hospital visits, and activity limitations are just some of the tricky aspects that must be faced, and that can expose children to higher risk of adjustment problems, such as anxiety [1,2,3]. Coping is a fundamental resource in this context. It can be defined as the intentional actions aimed at managing emotions, thoughts, behaviors, physiology, and the environment in response to challenging situations or events [4].

Delving into this aspect is an aim of high importance in the population of children with chronic diseases [5]. The literature reports that pediatric patients use a wide range of coping strategies, particularly social support-seeking, and avoidance [6,7,8]. Coping is related to children’s emotional adaptation: for example, some authors suggest that problem-focused coping and support-seeking strategies are more effective for the psychological well-being of children with chronic diseases, while avoidant coping strategies are less [9,10]. Thus, it is crucial to explore which are the most effective coping approaches and analyze their link to the psychological outcomes. Moreover, the last decades have seen a shift in attitudes toward childhood chronic illnesses: traditional models focused on deficits are gradually being replaced by models that highlight coping resources and the unique abilities of children and families [9].

Therefore, in the present paper, we chose to focus on coping, considering two specific chronic diseases: cancer and type 1 diabetes (T1D). These pathologies are among the most frequent pediatric chronic diseases and their prevalence is increasing worryingly [11,12]. They represent stressful conditions to cope with, on a spectrum of greater to lower life-threatening. Moreover, the two pathologies have already been compared in the literature, aiming to reach a better insight into the specific psychological implications of each chronic disease [9,13,14].

Several studies deal with parents’ coping strategies when facing their children’s cancer or diabetes, e.g., [15,16]; however, less is known about children’s strategies and how they relate to anxiety. Moreover, there is a scarcity of studies that have compared the coping mechanisms employed by children within these groups. Yet it is crucial to adopt an illness-specific approach to ensure appropriate care for these vulnerable populations [17].

When it comes to pediatric cancer, inconsistent results are reported by previous studies considering coping styles and psychological well-being. This may be because children with cancer show good psychological adjustment and normal scores in depression and anxiety, after a period of severe distress immediately after the diagnosis [18,19]. Moreover, the use of different assessment tools, timeframes, and stressful situations may account for the discrepancies in the results [1]. However, repressive coping styles seem to be characteristic of the population of children with cancer [18]. On the one hand, these strategies are linked to lower levels of children’s anxiety [20]; on the other hand, it has been shown that an avoidant coping style is associated with poorer adjustment, e.g., [21]. Disengagement or passive coping may not be effective in the context of low actual or perceived control levels of cancer. Furthermore, children who employ repression strategies tend to respond positively to questionnaires and often are unaware of their own psychological processes. It can be said that avoidance and distraction strategies are effective in the short term but more maladaptive in the long term [22].

Also, when it comes to children with diabetes, avoidance strategies are reported as the most used coping approaches [23]. Avoidant strategies have been connected to worse metabolic control and poorer psychological outcomes, such as lower quality of life and social competence, and greater depressive symptoms, while seeking social support and utilizing humor have been linked to better metabolic control [23,24,25]. The literature showed that coping strategies represent protective factors in adapting to T1D: problem-focused strategies are more useful for situations that can be modified, and emotion-focused coping in unchangeable ones [26]. Moreover, coping skills training interventions seem to be effective in the psychological adjustment to the disease and its management [24]. Still, much more must be explored when it comes to coping strategies in this population.

Linked to children’s coping strategies and adjustment is mothers’ anxiety. When it comes to pediatric cancer, the entire family system is affected by the child’s diagnosis, with objective and subjective difficulties, such as anxiety and depression. Usually, it happens that the mother is the primary caregiver; thus, they are more at risk for these bad consequences [27,28]. Higher levels of anxiety symptoms are also reported in the population of mothers of diabetic children: the daily duties and stressors linked to the disease, and the chronic nature of the illness are among the most evident sources of worries [29]. How mothers’ anxiety specifically interplays with children’s coping strategies in these populations is still unclear. It has been shown that children’s lower self-efficacy in managing disease is a predictor of greater caregiver burden [6]. However, to our knowledge, no studies are available about how children’s coping strategies and mothers’ anxiety relate to each other in pediatric cancer and diabetes.

This study aims to increase our understanding of the psychological outcomes elicited by a diagnosis of pediatric cancer or Type 1 diabetes (TD1) and, specifically, the impact of coping strategies on the severity of anxiety symptoms experienced by young patients and their primary caregivers. First, in line with previous studies, we expect to find differences between the clinical and control groups in the use of coping strategies. Coping is a multifaceted construct: in chronically ill children, the strategies are not necessarily activated against the illness but are employed to help the child manage and adapt to the daily challenges posed by the chronic condition (e.g., adapting to physical limitations during playtime) [10]. We aimed to assess the presence of differences in strategies used to cope with everyday life stressors in the context of healthy children and children with cancer or T1D. More specifically, we hypothesize to find greater levels of avoidant and distraction strategies in children with chronic diseases, compared to their healthy peers [18,23]. Considering healthy children, active coping seems to be the most used strategy, while avoidance and passivity are the least [30]. Moreover, we expected that, in the clinical samples, the use of coping strategies oriented toward seeking social support and proactive approaches might be associated with less severe anxiety in children and their mothers, while avoidance and distraction strategies might be linked to greater anxiety both in children and their caregivers [9,10]. The second more exploratory aim of the present study is to provide insights into possible differences in the management of severe anxiety outcomes associated with pediatric illnesses that should be tailored to the specific challenges that each diagnosis presents to both the patient and the caregiver.

## 2. Materials and Methods

### 2.1. Procedure

The present study has a cross-sectional design involving two groups of pediatric patients followed at the Regional Center for Pediatric Diabetes, at the University Hospital of Verona, and at the Pediatric Oncology Unit of Taranto (Italy), and their caregivers. Moreover, a control sample of children and their caregivers was recruited through snowball sampling, a chain-referral method.

The following inclusion criteria were applied for the clinical sample: age between 7 and 15 years old and a confirmed diagnosis of T1D or cancer; in this last case, at least two months needed to pass from the diagnosis, given the heavy stress that can occur immediately after [19]. The following were considered as exclusion criteria: the presence of other chronic or psychiatric diseases and poor comprehension of Italian. Regarding the control sample, inclusion criteria were age between 7 and 15 years, and exclusion criteria were a diagnosis of any pediatric chronic or psychiatric disease and scarce knowledge of Italian.

No reward was offered for enrolment. The project received approval from the Ethics Committee for Clinical Trials (CESC) (Observational study n. 977/CE) and by the Institutional Ethical Committee of Verona, Italy (Prot. n. 29,097). The research is in line with the Ethical and Deontological Codes of Italian Psychologists.

Pediatric patients were recruited by a researcher at the end of the control visits: children and their parents were introduced to the study and informed about the study design, method, and procedures. The pediatricians did not know who was willing to participate and who was not. In this way, freedom of choice was further preserved. Those willing to participate had to sign an informed consent. Children were verbally asked to give their agreement to be involved in the research and specific informant consent had to be signed for those between 12 and 15 years of age.

Data were collected between November 2020 and May 2021. The group of children with T1D and their caregivers completed the questionnaires in person, in a separate and quiet room of the University Hospital of Verona, under the supervision of the ward’s psychotherapist, after the routine visit, while the group with cancer filled out the surveys online, using a link to a Google form, that was sent by email. They completed the questionnaires from home. Data confidentiality was guaranteed. Data collection was planned to not interfere with the medical procedures and to respect the COVID-19 safety guidelines. The timing of the control sample’s data collection was the same as the clinical samples: caregivers were contacted by phone and informed about the study by the researchers; they gave verbally their consent for participation. Then, appointments in person were planned to sign the informed consent and complete the questionnaires, following the ethical guidelines. The surveys and questionnaires were equivalent, except for the items regarding pediatric diseases. For each subgroup, between 91% and 97% of individuals who met the inclusion and exclusion criteria consented to participate in the study. More specifically, 3 families of children with cancer and 2 families of children with diabetes refused to take part in the research. Reasons for refusal included time constraints and lack of interest in the research. In addition, children with cancer were 30–40% of those followed by the Pediatric Unit at Taranto Hospital, while children with diabetes represented 55% of those followed by the Regional Center for Pediatric Diabetes, at the University Hospital of Verona (out of the total number of children aged 0 to 21 years).

### 2.2. Measures

The interview consisted of a survey created ad hoc and some standardized questionnaires.

As for caregivers, questions were related to them and their children: sociodemographic information (e.g., age, occupation) and clinical data about their children’s disease (e.g., time since the diagnosis) was sought. Moreover, they filled in standardized questionnaires.

The General Health Questionnaire-12 (GHQ-12) [31] is a 12-item, self-report screening instrument to assess short-term psychological well-being and, in particular, anxiety and depressive symptoms in the past 2 weeks. Each item is evaluated on a 4-point Likert scale, from 0 “more than usual” to 3 “much less than usual”. Cut-off scores distinguish three ranges: scores 0 to 14 correspond to normal global functioning, scores 15 to 19 indicate psychological suffering, and scores 19 to 36 indicate significant distress. The validity and reliability of the Italian version have been confirmed [32]. In this study, Cronbach’s alpha was 0.83.

The State-Trait Anxiety Inventory-Y [33] is a self-report tool assessing anxiety in its components: state and trait anxiety. The first one evaluates current anxiety symptoms, while the second one shows the tendency to react anxiously to a perceived threat. The questionnaire consists of 40 items, 20 for the state and 20 for the trait anxiety scale. Items are rated on a 4-point Likert scale from 0 “almost never” to 3 “almost always”. The Italian version of the STAI-Y has been validated [34]. In the present study, Cronbach’s alpha was 0.96.

Children had to fill in standardized questionnaires.

The Strengths and Difficulties Questionnaire (SDQ) [35] is a brief behavioral screening questionnaire to assess the psychological adjustment of children and adolescents, consisting of 25 items, rated on a 3-point Likert scale, from 0 “not true” to 2 “certainly true”. It uses five scales: Emotional Symptoms (sdq_emo), Conduct Problems (sdq_cond), Hyperactivity–Inattention (sdq_hyper), Peer Problems (sdq_peer), and Prosocial Behavior (sdq_pros). Summing the first 4 scales, the Total Difficulties Score (sdq_tds) is obtained; scores 20 and up indicate the clinical range. The questionnaire has been validated for children and adolescents from 8 to 18 years old [36]. In this study, Cronbach’s α for the total difficulties score and the subscales emotional symptoms, conduct problems, hyperactivity–inattention, peer problems, and prosocial behavior were, respectively, α(TDS) = 0.81, α(emo) = 0.72, α(con) = 0.52, α(hyp) = 0.63, α(peer) = 0.60, and α(pros) = 0.72.

The Children’s Coping Strategies Checklist (CCSC) [37] is a 54-item self-report tool for children and adolescents, which assesses the construct of coping and the individual’s utilization of various strategies. The calculated factors are indeed five: “Problem-focused coping”, “Positive cognitive restructuring”, “Distraction strategies”, “Avoidance strategies”, and “Support-seeking strategies.” Each item is assessed on a 4-point Likert scale ranging from “never” to “most of the time”. The Italian version has been validated [38]. In the present study, Cronbach’s alphas for the subscales Problem-focused coping, Positive cognitive restructuring, Distraction strategies, Avoidance strategies, and Support-seeking strategies were, respectively, α(PROBFC) = 0.83, α(POSCOGR) = 0.76, α(DISSTRA) = 0.76, α(AVOSTRAT) = 0.59, and α(SUPSS) = 0.81.

The State-Trait Anxiety Inventory for Children (STAI-C) [39] is an adequate and specific tool that assesses anxiety symptoms in childhood. As for the adult version, it has 2 scales: the state and trait anxiety. Children have to rate 40 items, 10 for each scale. The Likert scale ranges from 0 “almost never” to 3 “almost always”. The Italian version has been validated [40]. In the present study, Cronbach’s alpha was 0.80.

### 2.3. Statistical Analysis

To analyze the current rich dataset, we have chosen two primary approaches. The first is a descriptive approach aimed at clarifying the sample’s characteristics and visualizing the data distribution. Visualizing descriptives and distributions in psychology studies is crucial for gaining insights into the underlying patterns and characteristics of data. It enables researchers to identify trends, outliers, and potential relationships, providing a foundational understanding that informs subsequent statistical analyses [41]. Our visual exploration not only enhances the interpretability of results but also informs hypothesis testing, ultimately contributing to the robustness and informativity of the results. The second is the inferential approach; specifically, we selected multiple linear regression models to statistically estimate the effects linking patient coping styles across the three groups (control, cancer, and T1D) to anxiety levels measured in the same patients and their mothers with a multilevel multiple regression analysis approach. To perform and visualize the result analysis, we utilized R software version 4.2.1 [42] along with the lme4 package [43] and SjPlot [44].

## 3. Results

Two clinical and one control sample of 7 to-15-year-old children and their parents were collected for the present research, for a total of 150 participants (see Table 1 for demographics). Of these, fifty-six children had a diagnosis of T1D, thirty-three of cancer, and sixty-one were healthy. No significant difference was found between the groups as to age (*p* = 0.688).

When it comes to the first group, 21 were females and 35 were males. The mean age for children was 11.64 (S.D. = 2.46 years). They were recruited among those regularly followed by the Regional Center for Pediatric Diabetes, at the University Hospital of Verona (Italy). The diagnosis of T1D was made on average 67.14 months before the collection time (S.D. = 40.59). The mean percentage of time with in-range glycemic levels was 60.42% (%TIR). In total, 98.1% of the subgroup scored in the normal range as to psychological adjustment (SDQ total score). Also, their caregivers participated in the study: their mean age was 44.05 (S.D. = 5.74) and they were mainly full-time workers (38.9%). A total of 16.1% of them reported significant psychological distress, as to their general well-being.

The group of children with cancer was composed of 17 females and 16 males, and the mean age was 11.12 (S.D. = 3.15). The diagnosis was made on average 22.32 months before the collection time (S.D. = 12.36). The following percentages describe the type of cancer diagnosed: 45.5% hematologic malignancy, 39.4% solid tumor, and 15.2% other hematological pathologies. A total of 93.3% of the subgroup scored in the normal range as to psychological adjustment (SDQ total score). Also, their caregivers participated in the study: their mean age was 41.76 (S.D. = 6.21) and 43.1% of them were housewives/unemployed. A total of 33.3% of them reported significant psychological distress, as to their general well-being.

The group of healthy children (control group) was composed of 27 females and 32 males, and the mean age was 11.36 (S.D. = 2.52). A total of 89.5% of the subgroup scored in the normal range as to psychological adjustment (SDQ total score). Also, their caregivers were involved in the study: their mean age was 44.97 (S.D. = 5.43). The most observed category (42.6%) was part-time workers. As to their general well-being, 23% of them scored in the range indicating significant psychological distress.

Taking a look at the coping strategy distributions observed in the three investigated samples, as shown in Figure 1, it emerges that the frequency at which the five investigated coping strategies occur in our data collection is similar among the three groups. They all exhibit a lower tendency to adopt distraction and support-seeking strategies compared to the other coping strategies. This aspect is in contrast with our expectations, yet is relevant in the context of our investigation by suggesting that the three groups are comparable in terms of coping strategies. This allows us to estimate the interactive effect between coping strategy and diagnosis on the anxiety levels of both patients and their caregivers.

To explore the impact of children’s coping strategies on the anxiety levels of both children and their mothers, we first computed the degree of overlap between the anxiety levels of mothers and children in the three groups, as shown in Figure 2 [45]. This approach enables us to derive a statistical index regarding the degree of overlap between caregivers and children’s anxiety in a more precise manner, taking into account the sample size and, most importantly, minimizing the influence of outliers in determining whether the two are different. Figure 2 displays graphs depicting the overlap of the distribution scores provided by children and parents. The overlapping index is 0.68 for the control group, 0.47 for the cancer group, and 0.43 for the TD1 group. Please note that a value close to 0 indicates completely separate distributions, implying that the two groups differ, whereas a value close to 1 suggests significant overlap, making it difficult to differentiate between the two distributions.

Following the logic of the present data analysis, we tested the interactive effect of diagnosis (i.e., the three groups) and coping strategies on the anxiety levels reported by both children and mothers for each of the five coping strategies under examination: “Problem-focused coping”, “Positive cognitive restructuring”, “Distraction strategies”, “Avoidance strategies”, and “Support-seeking strategies”.

### 3.1. Problem-Focused Coping

As shown in Figure 3 we did not find any interaction effect between the use of problem-focused coping strategies on the anxiety levels measured with the STAI in children (b = −0.25, SE = 0.43, T = −0.59, *p* = 0.55) and their caregivers (b = 1.11, SE = 0.70, t = 1.57, *p* = 0.118). It is worth noting the opposite numerical trends found in Figure 3a. the caregivers’ vs. Figure 3b. children’s STAI depending on the frequency of the problem-focused coping strategy in the cancer group.

### 3.2. Positive Cognitive Restructuring

As shown in Figure 4, we found a substantial group effect, with the caregivers of the T1D group showing higher levels of anxiety compared to the control group (b = 32.12, SE = 15.55, t = 2.07, *p* = 0.040), and a substantial interaction indicating that a higher frequency of positive cognitive restructuring is associated with higher levels of anxiety in the caregivers of the cancer group (b = 1.46, SE = 0.68, t = 2.16, *p* = 0.03). No effect of the positive cognitive restructuring strategy and the group emerged on the STAI measured in children (b = 0.03, SE = 0.41, t = 0.08, *p* = 0.94).

### 3.3. Distraction Strategies

As shown in Figure 5, we did not find any effect of distraction strategies and groups on the anxiety levels of caregivers (b = 0.09, SE = 0.66, t = 0.14, *p* = 0.88). However, we found a substantial interaction between group and distraction; that is, a decrease in anxiety levels in the cancer group and an increase in the anxiety levels in the control group in those children that more frequently use distraction as a coping strategy (b = −1.01, SE = 0.40, t = −2.54, *p* = 0.01).

### 3.4. Avoidance Strategies

As shown in Figure 6, we did not find any substantial effect of the avoidance strategies and the group on the anxiety levels measured in the children (b = −0.45, SE = 0.49, t = −0.92, *p* = 0.36) and their caregivers (b = 1.23, SE = 0.75, t = 1.63, *p* = 0.10). It is worth noting the opposite numerical trends found in the caregivers’ vs. children’s STAI depending on the frequency of the avoidance strategies, in the cancer group.

### 3.5. Support-Seeking Strategies

As shown in Figure 7, we did not find any interactive effect between support-seeking strategies and groups on anxiety levels measured with the STAI in caregivers (b = 0.99, SE = 0.77, t = 1.28, *p* = 0.20) and their children (b = −0.04, SE = 0.50, t = −0.08, *p* = 0.94).

## 4. Discussion

The present study aims to enhance our comprehension of the psychological outcomes that result from a diagnosis of pediatric cancer or T1D, with a specific focus on how coping strategies influence the severity of anxiety symptoms in young patients and their primary caregivers. It is worth noting that the importance of this study also lies in the fact that we are comparing these complex family dynamics in response to diagnoses of cancer and diabetes, which present very different challenges and scenarios in the management of care over the medium and long term.

First, the nuanced depiction of distribution densities, both in coping strategies and anxiety levels (i.e., STAI, Figure 1 and Figure 2), allowed us to paint a vivid picture of the dynamics at play. Specifically, the visualization of response densities pointed out that children could be compared based on their preferred coping strategies regardless of the specific group. They all showed lower adoption of distraction and support-seeking strategies compared to the other coping mechanisms, suggesting that not all the strategies are equally used. This result does not align with the hypothesis and the literature [18,23,30]. However, it has to be said that most studies on children’s coping strategies fail to reach situational sensitivity, making it challenging to compare the results [46]. The CCSC does not specifically assess reaction to disease-related stressors, concerning which differences may be more evident. For instance, one study discovered similarities in the coping strategies employed when dealing with common stressors, comparing groups of children with chronic diseases to their healthy peers [46]. Coping is an “organizational construct” used to encompass the multitude of actions individuals use to cope with stressful situations: it is related to a variety of variables such as the type of stressor, the context (e.g., school, family), sociodemographic and economic factors, and past stressing demands [38]. That considered, it seems that preferred coping strategies against everyday life stressors are similar in the context of healthy children and those with T1D and cancer. However, this result increased the validity of the investigation of the impact of coping strategies on anxiety levels reported by both caregivers and their children among groups. In particular, the richness of data visualization showed that distraction and support-seeking coping were consistently utilized less across all groups.

Nevertheless, the results highlighted a statistically relevant different effect of distraction on children’s anxiety levels. In contrast with our hypothesis, distraction exhibited an immunizing effect on anxiety levels within the group of children diagnosed with cancer, yet was associated with a cost in terms of increased anxiety levels within the control group. The existing literature showed disengagement or passive coping not to be effective strategies in the context of low actual or perceived control levels of many pediatric chronic diseases. Emotional detachment, denial, or avoidance, for example, do not help effectively regulate emotional distress or directly confronting the stressful situation, and may constitute an obstacle to other coping strategies aimed at adjusting to uncontrollable stress [10]. It can be argued that avoidance and distraction strategies may prove effective in the short term, but they become less adaptive over an extended period [22]. The children in the current study sample received their diagnoses, on average, two years before data collection. This might be considered a timeframe in which the distraction strategies still have beneficial effects. Moreover, Phipps and Steele [20], for example, found that repressive coping styles in children with cancer are connected to lower children’s anxiety symptoms. Another way of interpreting the results may be that when confronted with life-threatening events, such as cancer, suppressing or restricting awareness of stimuli that could trigger anxiety may be beneficial [19]. Indeed, cancer is more life-threatening than diabetes, which can be more easily and effectively managed. Adolescents who employ repressive strategies, for example, appear to have a higher tolerance for frustration, better academic performance, and improved social skills compared to non-repressors [47]. The role of distraction as a coping strategy showed its potential to mitigate anxiety in the context of illness, while potentially exacerbating anxiety in different circumstances.

Interestingly, the study did not identify an impact of support-seeking strategies on anxiety levels in either group, while we expected a positive role of this coping style in mitigating children’s anxiety in the clinical groups [9]. Various factors may have played a role in these findings. For instance, the quality of social support received, both within the family and in other contexts, might have influenced the responses. Moreover, the wide age range in our sample should be mentioned, with younger children potentially placing more emphasis on seeking social support compared to adolescents [48]. Additionally, the effectiveness of specific strategies may vary depending on the stage of the disease; for example, strategies employed in the initial phases might differ from those in later periods. Furthermore, it is crucial to acknowledge that perceived social support and anxiety management can be influenced by individual psychological factors, such as resilience, self-esteem, and perceived control. These hypotheses require further exploration, suggesting that while these two strategies are not prevalent choices among children, their utilization drives a distinguishable influence on children’s anxiety levels.

Moreover, the examination of anxiety levels reported by caregivers and their children, analyzed through the lens of the overlapping index, indicated that despite caregivers generally exhibiting higher anxiety levels compared to their children across all groups, the control group showed a greater degree of overlap between mothers and children. Moreover, this group, representing a non-clinical sample, presented lower anxiety levels in comparison to the two clinical cohorts. This observation sparks thoughtful considerations regarding the interplay of anxiety within familial dynamics and the impact of a clinical diagnosis on this interplay [49]. The increased anxiety experienced by caregivers in the clinical samples underscores the burden often shouldered by parents or caregivers in such circumstances [50]. However, the increased overlap in the control group provides a unique insight. It suggests a closer tuning of anxiety experiences between mothers and children in non-clinical settings. As the literature reports, in healthy children’s populations, maternal anxiety disorders are associated with anxiety disorders in their children [51].

In addition, the statistical regression analysis explored the impact of diagnosis and coping strategies on children’s anxiety levels, as well as those of their mothers. We did not find any significant impact of problem-focused and avoidance coping strategies. However, by considering the direction of the effect estimated by regression models, there was a noteworthy trend indicating a positive association between higher anxiety levels in caregivers and lower anxiety levels in children within the group diagnosed with cancer as the utilization of problem-focused and avoidance strategies increased, as shown in Figure 3 and Figure 6. To the best of our knowledge, no studies have investigated the influence of coping strategies employed by children with cancer or diabetes on their mothers’ anxiety. This intriguing trend needs further investigation, as it can reveal specific dynamics of anxiety within the family unit that might explain the lower level of overlap between mothers and children in the clinical sample. It suggests that caregivers experience increased anxiety levels, while, conversely, children demonstrate lower anxiety levels when they utilize both problem-focused and avoidance coping strategies. It seems that problem-focused and avoidance coping strategies are effective for children who use them in the cancer group but have a negative impact in terms of anxiety on their mothers. This finding prompts a closer examination of the knowledge and awareness concerning the effectiveness and costs associated with coping strategies among caregivers.

Moreover, according to the interpretation of the data discussed thus far, we found an opposing effect of Positive Cognitive Restructuring coping strategies on anxiety levels in the group of caregivers with children diagnosed with T1D compared to those with children diagnosed with cancer. In the former, an increase in children’s utilization of positive cognitive restructuring was associated with a reduction in caregivers’ anxiety levels, while conversely, a statistically significant increase in anxiety levels was observed in the group of caregivers with children diagnosed with cancer. It seems that children’s use of positive cognitive restructuring is beneficial for caregivers in the T1D group, while it is detrimental for those in the cancer group. In the present study, children with cancer received the diagnosis on average 2 years before the data collection, significantly more recently than the T1D groups. Children with cancer tend to show good psychological adjustment and normal levels of anxiety, after a period of severe distress immediately after the diagnosis [18,19]. This opposite effect of the positive cognitive restructuring coping strategies on anxiety levels underscores the necessity of tailoring interventions based on the specific disease context. It further emphasizes the need for targeted psychoeducational interventions, focusing on children’s coping strategies, and also engaging caregivers. These interventions should be carefully designed to suit the short-term and long-term management requirements of diseases with vastly different temporal characteristics. Moreover, integrating the observed trends concerning problem-focused and avoidance strategies with these data and considering the evident discrepancy in anxiety levels (manifested as low levels of overlap) observed in clinical groups, the present study underscores the imperative for disease-specific coping strategy psychoeducation interventions.

Lastly, it is interesting to note that, even if not statistically significant, in the group of children with cancer, the use of all coping strategies is linked to a trend of increased caregivers’ anxiety, but in caregivers of children with diabetes a (lower) trend for decrease. The numerical trends may be due to the time passed since the diagnosis (which differs in the two groups), or to the burden of the disease itself, considering the less life-threatening condition of T1D. However, when it comes to children of both clinical groups, a trend toward decreased anxiety can be noted in association with the use of all coping strategies, even if not always statistically significant. It may be thought that children develop a certain degree of competence in coping strategies when dealing with a chronic disease and that the use of just one of them is beneficial for their well-being. The topic needs further exploration: a better understanding of the components of each strategy may be helpful to better design interventions.

The current study has several limitations. Firstly, the CCSC does not specifically assess coping strategies concerning disease-related stressors, thus influencing our results. Additionally, the group of children with cancer received their diagnosis significantly more recently than the T1D group. Furthermore, it is worth noting that data were collected in different settings for the two clinical groups: children with T1D and their caregivers completed the questionnaires in person at the hospital, while the group with cancer filled out the surveys online. The control sample’s data were also collected in person. Also, the wide age range has to be mentioned: children may use different coping strategies based on their developmental stage. That is, a punctual investigation of the specific disease phase may have been conducted, as it may influence the coping strategies used and their impact on anxiety levels of both children and their caregivers. Lastly, we included children aged 7 years in the present study’s samples, although the questionnaires are validated for older ages. However, it has to be mentioned that they have been used in previous studies with children as young as 7 [38,52,53,54].

Future research might consider longitudinal designs to better understand how coping strategies change over time after the diagnosis of different diseases. Moreover, employing more specific tools to evaluate how children with chronic diseases and their parents cope with various kinds of stressors and in specific contexts, focusing specifically on illness-related stressors, could be beneficial. Further investigations could be conducted to explore how coping strategies vary with age and gender in chronically ill children’s populations. It would also be interesting to assess the coping strategies of parents and their impact on the psychological well-being of both them and their children. Special attention could be given to the role of fathers in this context. Moreover, parenting style plays a crucial role in children’s mental health and their selection of coping strategies: future research may consider this variable when it comes to coping. Additionally, studying the interplay between children and their caregivers’ employed coping strategies could provide valuable insights into the subject. Lastly, the same research questions on coping strategies and their interaction with children’s and parents’ anxiety may involve other vulnerable populations such as children with neuropsychiatric disorders and their families [55].

## 5. Conclusions

The study moved a step forward in the intricate exploration of coping strategies employed by children and how these strategies resonate with their mothers, shedding light on their mutual well-being. Considering the first hypothesis, it emerged that the three groups of children (with cancer, T1D, and healthy) were comparable in the preferred coping strategies. Moreover, in contrast with the hypothesis, distraction strategies were found to have an immunizing effect on children’s anxiety within the group of children diagnosed with cancer. In the same group, greater adoption of problem-focused or avoidant coping by the children was linked to greater caregivers’ anxiety, while an opposing effect of Positive Cognitive Restructuring coping strategies was found on anxiety levels in the group of caregivers of children with T1D. In particular, the visual representation of data distribution enriched our understanding of the diagnosis’s influence on both coping strategies and anxiety levels among family actors. The assessment of anxiety overlapping within the caregiver–child dyad, differentiated by clinical context, offers a valuable lens through which we comprehend the intricate relationships between anxiety, familial roles, and the influence of clinical circumstances. The differences in anxiety levels between clinical and non-clinical groups emphasize the critical role of diagnosis and its effects on family well-being. The higher anxiety levels within the clinical groups indicated a potentially destabilizing effect of a diagnosis on anxiety levels across family members. Even when not statistically significant, the trends identified by the study provide valuable information to be further investigated. Using a theoretically driven data exploration, the present study capitalized on a rich palette of information that extends beyond the confines of statistical significance. The proposed approach not only reveals the complexity of the phenomenon under investigation but also needs the transparent sharing of methodological challenges and solutions. The present study underscores the necessity of considering these nuanced dynamics in tailoring interventions that address the needs of both children and caregivers. By focusing on educating and supporting caregivers regarding effective coping strategies, we can develop more tailored interventions that enhance the overall well-being of families dealing with the challenges of a specific diagnosis. For example, it may be useful to work on Positive Cognitive Restructuring with children diagnosed with T1D, as a greater use of the strategy seems to be associated with a reduced caregivers’ anxiety. In essence, recognizing the variability in coping strategy effectiveness within different medical contexts is pivotal. By understanding the unique coping needs associated with various medical conditions, we can develop targeted strategies to enhance the psychological well-being of both caregivers and children, optimizing the management of diverse health conditions.

## Figures and Tables

**Figure 1 ijerph-21-00077-f001:**
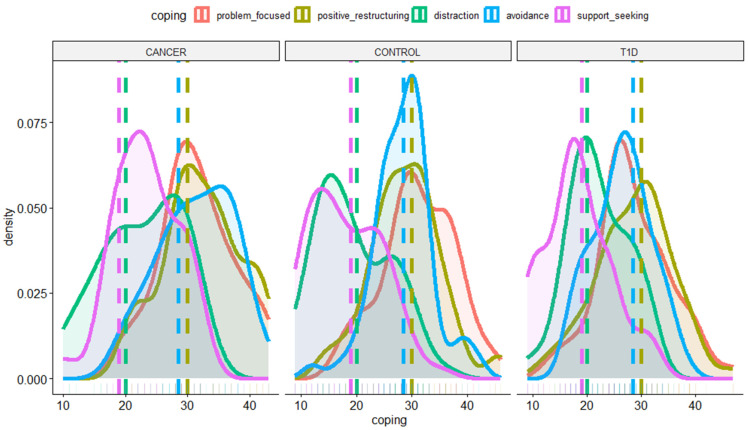
Distribution of values of the coping strategies (problem-focused, positive cognitive restructuring, distraction, avoidance, and support-seeking) captured in the three groups (Cancer, Control, and T1D). Vertical lines indicate the median value for each distribution.

**Figure 2 ijerph-21-00077-f002:**
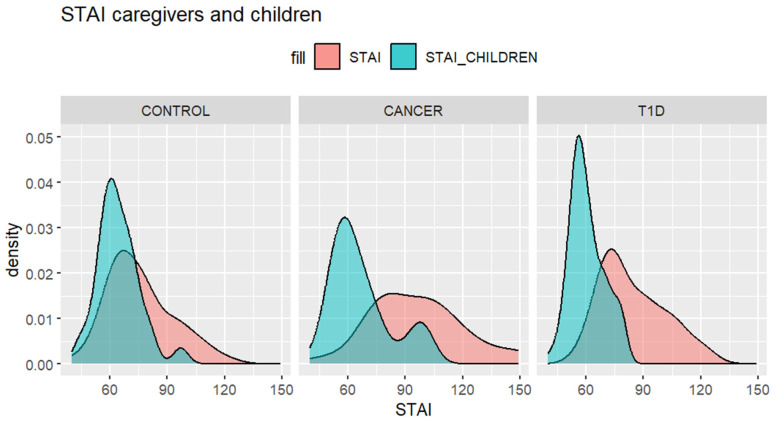
Density estimates of the three groups and overlap area.

**Figure 3 ijerph-21-00077-f003:**
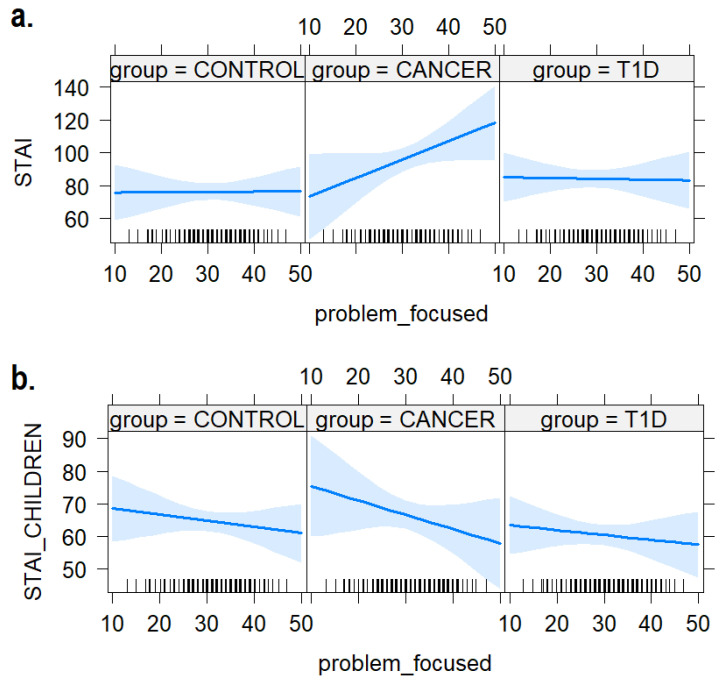
The plot of the interaction effects between children’s use of problem-focused coping skills and anxiety levels in (**a**) mothers and (**b**) the children themselves. The areas around the regression line represent the 95% confidence intervals.

**Figure 4 ijerph-21-00077-f004:**
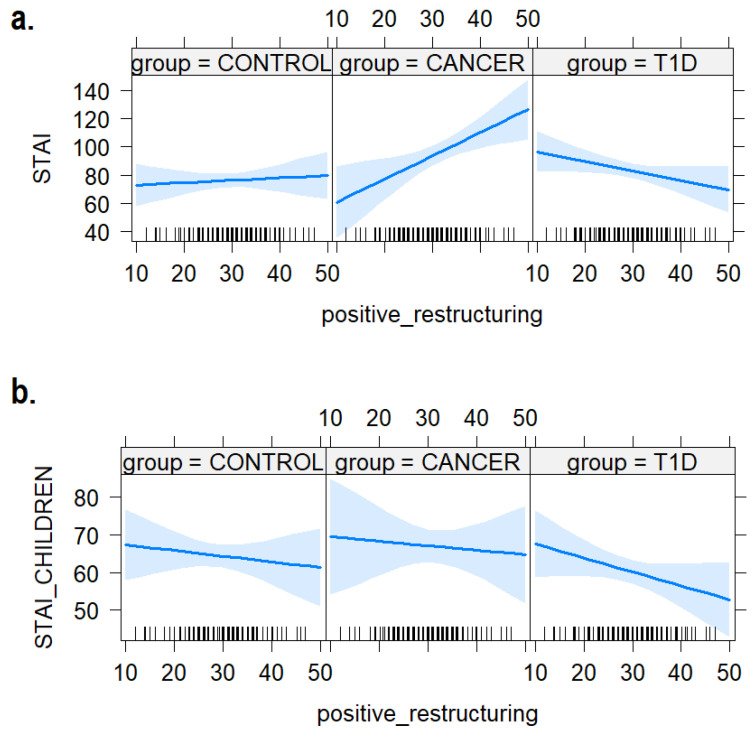
The plot of the interaction effects between children’s use of positive cognitive restructuring coping skills and anxiety levels in (**a**) mothers and (**b**) the children themselves. The areas around the regression line represent the 95% confidence intervals.

**Figure 5 ijerph-21-00077-f005:**
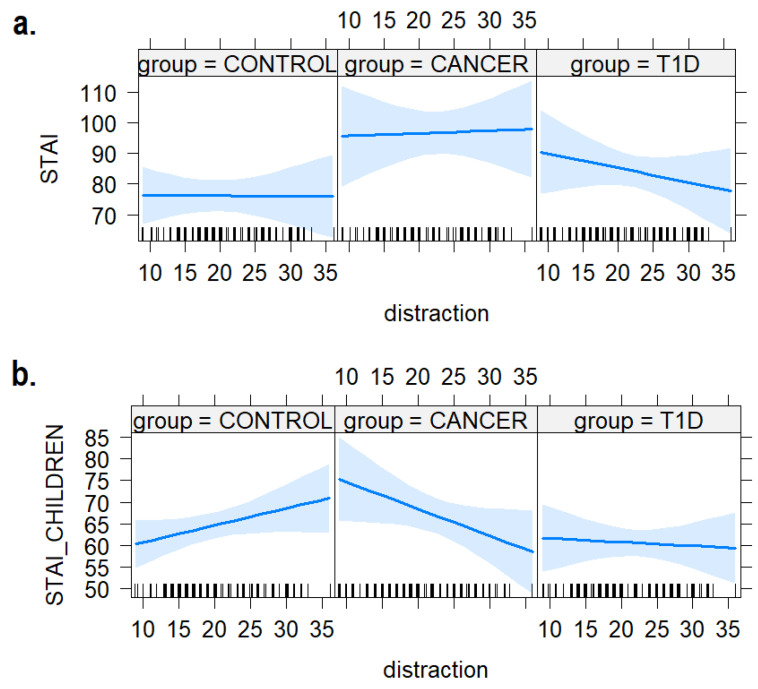
The plot of the interaction effects between children’s use of distraction strategies and anxiety levels in (**a**) mothers and (**b**) the children themselves. The areas around the regression line represent the 95% confidence intervals.

**Figure 6 ijerph-21-00077-f006:**
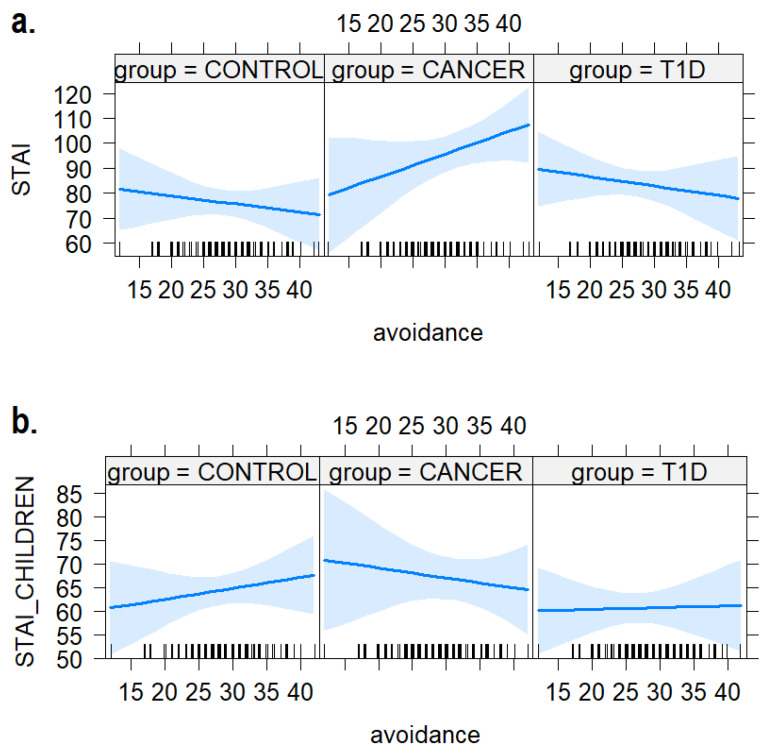
The plot of the interaction effects between children’s use of avoidance strategies and anxiety levels in (**a**) mothers and (**b**) the children themselves. The areas around the regression line represent the 95% confidence intervals.

**Figure 7 ijerph-21-00077-f007:**
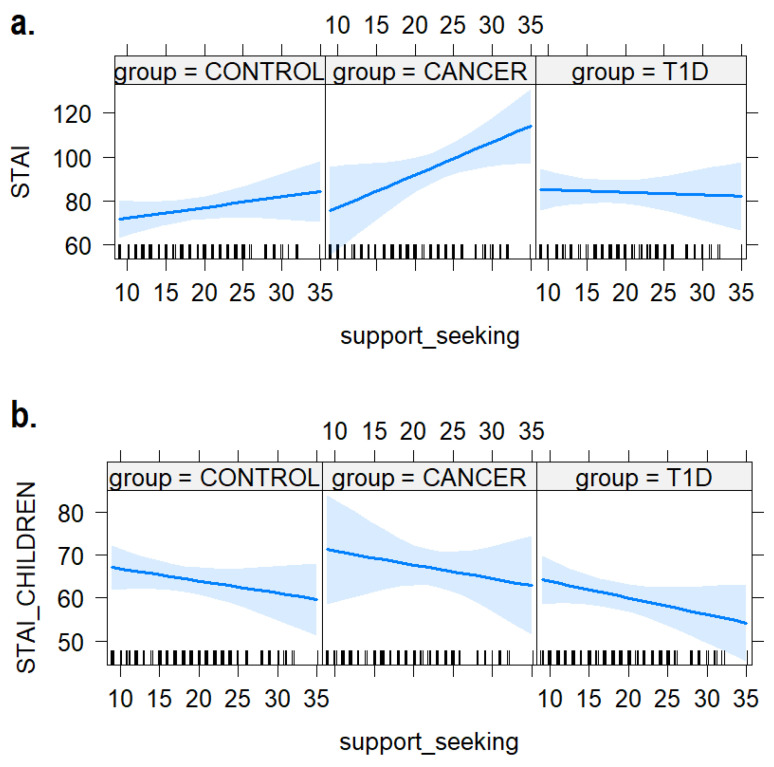
The plot of the interaction effects between children’s use of support-seeking strategies and anxiety levels in (**a**) mothers and (**b**) the children themselves. The areas around the regression line represent the 95% confidence intervals.

**Table 1 ijerph-21-00077-t001:** Sociodemographics for the clinical and control samples.

Children	Control (*n* = 61)		Cancer (*n* = 33)		T1D (*n* = 56)	
	Mean	Sd	Mean	Sd	Mean	Sd
Age	11.36	2.52	11.13	3.15	11.64	2.46
Gender	32 M/27 F		16 M/17 F		35 M/21 F	
Time from the diagnosis (in months)			22.32	12.36	67.14	40.59
%TIR					7.28	0.75
Caregivers	Control (*n* = 61)		Cancer (*n* = 33)		T1D (*n* = 56)	
	Mean	Sd	Mean	Sd	Mean	Sd
Age	44.97	5.43	41.76	6.21	44.05	5.74
Occupation						
Full-time worker	*N* = 24 (39.4%)		*N* = 7 (21.9%)		*N* = 27 (38.9%)	
Part-time worker	*N* = 26 (42.6%)		*N* = 8 (25%)		*N* = 13 (23.2%)	
Housewife/unemployed	*N* = 11 (18%)		*N* = 17 (43.1%)		*N* = 16 (28.6%)	

## Data Availability

The data presented in this study are available on request from the corresponding author. The data are not publicly available because they report private information about participants.

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
