# Peer review of "Examining Coping Strategies and Their Relation with Anxiety: Implications for Children Diagnosed with Cancer or Type 1 Diabetes and Their Caregivers"

_ijerph, 2024, doi:10.3390/ijerph21010077_

Round 1
Reviewer 1 Report
Comments and Suggestions for Authors
The article is original, relevant, well-justified, and well-framed.
The summary described in the study, but the conclusions and suggestions could be better in relation to alignment with the objectives.
Introduction Section: The study is well theoretically framed, although the vast majority of references are more than 5 years old.
Materials and Methods Section: The research design appears appropriate. However the methodology can be improved in the description of the type of study, as it only refers to a study that has a cross-sectional design and in the application of the instruments with regard to completion times, the way in which the elements of the control group were recruited and the location of completion.
Results Section: The results are well presented, but for better understanding, I suggest that the statistical measures Mean (n, Percentage) from Table 1 be placed next to the variables. Ex: Age, mean or full-time worker, n (Percentage), and the figure titles are shorter
Discussion Section: Throughout the text, the description of the acronym (T1D) in lines 331, 404, and 428 must be eliminated, as it already occurred in line 53. In the paragraph that describes the limitations, you can include references and justify the fact that children aged 7 years were included and that most of the instruments used are validated for older ages.
Conclusion Section: Your conclusions aptly encapsulate your findings. It might be worth mentioning clearer alignment with objectives.
Review the bibliographic list, there are some references without DOI. Ex: 34, 37, ...
Reviewer 2 Report
Comments and Suggestions for Authors
I have assessed the manuscript “Examining Coping Strategies and Their Relation with Anxiety: 2 Implications for Children Diagnosed with Cancer or Type1 3 Diabetes and Their Caregivers”. I thoroughly enjoyed reading the article, which presented a well-analyzed and explored topic. I have a few suggestions to enhance some of the concepts in the manuscript, but overall, it was a great read.
The theoretical review is based on sound foundations and effectively discusses coping mechanisms and anxiety associated with chronic childhood illness. It also highlights the significance of evaluating psychological factors in children and their caregivers, as well as the fluidity of these relationships within a family unit.
The objective is clear, as are its hypotheses. However, when we read the whole article, we see that these hypotheses are not upheld.
There are some aspects of the method that require clarification. Specifically, the recruitment of participants, including the control group, needs to be more detailed. While the inclusion criteria are explicit and the authors explain that the children were selected after medical consultations, it is unclear in the method how many families declined to take part. The refusal rate is mentioned in the results, but it should also be included in the method, along with more comprehensive information on the recruitment process. The families of children with cancer and children with diabetes represent what percentage of the cases seen at the hospital? Additionally, the recruitment and data collection process of the control group should be outlined to give the reader an idea of how it was conducted. It is also imperative to specify the duration (in months) during which the data was gathered.
Regarding the results, a question arises concerning coping: did the clinical group provide information about the coping mechanisms they utilise to manage their illness when answering the instrument? Furthermore, how does this compare to the control group? The convergence of the three groups' responses on coping suggests an inherent issue that pertains to the dynamic nature of the concept of coping. This must be examined within a specific context. So, the query is: is it logical to compare the coping mechanisms of ill children with those of healthy children?
The results regarding the low levels of distraction are interesting and positive, as they also reveal that there is a diversity of strategies used by the participants.
In Figure 1, differentiating between the distraction and avoidance data is challenging due to the closely related colours.
The findings indicating no interaction between problem-focused coping strategies and anxiety levels are interesting, as they show that emotion does not go directly into behaviour. The decrease in anxiety levels in the cancer group and the increase in anxiety levels in the control group merit further exploration.
In the discussion, I suggest that the authors provide further elaboration on their findings that there are no variances in coping methods among the three groups. How can we explain the role of distraction in alleviating anxiety in children with cancer? The authors clarify the significance of distraction in the context of health by emphasizing the ability for children to overcome painful and fearful medical procedures. However, why did the authors initially regard distraction to be an unproductive strategy?
The minor role of support-seeking strategies is a surprise. How can the authors interpret this result?
In summary, the paper addresses a quite important topic to discuss and to advance in the coping with pediatric chronic illness field. Sao necessarias alguns pequenos ajustes para sua publicaçao. A few minor adjustments are required for publication.
Reviewer 3 Report
Comments and Suggestions for Authors
The study titled “Examining Coping Strategies and Their Relation with Anxiety: Implications for Children Diagnosed with Cancer or Type1 3 Diabetes and Their Caregivers” is very interesting and rather bold for performing the analysis health in two chronical diseases considering two populations (child and caregivers). I agree with the limmitations discussed by the authors (i.e. different time lapse since the diagnoses for the diseases and different ways of collecting data) however I consider they provide more insights for future research refinements than invalidates the data collected. Although the no significant relation of the studied variables was identified I like to think a good research is not only those with robust results but also those that offer good insights for the field. That said, I recommend the study for publication since its methods are clear and adequate and the analysis and discussion are compatible to the data. However, I strongly recommend the authors include some extra considerations on the discussion which would enhance the paper relevance. As mentioned by the authors: “Visualizing descriptives and distributions in psychology studies is crucial for gaining insights into the underlying patterns and characteristics of data. It enables researchers to identify trends, outliers, and potential relationships, providing a foundational understanding that informs subsequent statistical analyses” , I would add that even when not statisticaly significant theses trends provide valuable information to be further investigated. Taking it into account please consider:
In Figure 1: the differences in control and experimental groups is very interesting, it indicates that the caregiver-children dynamic is directly affected in a way that deserver specific investigations. It can be observed that when anxiety of child with chronic disease is higher caregiver anxiety is lower (left part of the graph), do parents change their coping strategies when kids are more distressed? Do parents suppress their own emotion in this situation? Anyway, it shows that mental health of caregivers and children, in relation to dealing with chronic disease have distinct factors and would deserve differential attention. It would be interesting to give it some more lines of discussion.
In figure 2: it basically informs that chronical disease does not affect the coping strategies of children, more specifically does not push coping to different distributions. Coping strategies pattern are very complex, they are affected by actual demands (stressors), past stressing demands, social-economical factors, individual repertoire, quality of personal support, and such… I think this deservers a few comments, otherwise it would seem they are static personality traits.
Figures 4-6: Those are the ones I would recommend some more consideration for the descriptive (visual) trends:
1) Although generally not significant, it is interesting to notice that for cancers caregivers, in all coping strategy there is a trend of anxiety increase, and for diabetes caregivers (in a much lower level) a trend for decrease. Maybe this is due to the time-lapse of diagnose, or for the burden of the disease itself (different study designs could address this question).
2) in as opposite direction for children there seems to be a general trend for both experimental conditions, that every coping strategy (in different levels) is associated to a decrease in anxiety. Again it is important not to take a slight trend as a probable cause, but as it is observed consistently it deserves some attention. Is it possible that there are degrees of “competence” on the strategy and that all of them could have beneficial effects in dealing with chronical diseases? If so, we would have to understand better the components of each strategy to better design coping interventions.
I strongly agree with the authors when they mention that further studies should assess caregiver´s coping strategy, maybe the combination of caregiver and child strategy would be rather important for understanding mutual anxiety. I would further suggest that parental style would be assessed, parents may be more authoritative, repressive, permissive, negligent, loving, supporting and so on, there are standard scales for this evaluation ant they may play important role in mental health of the child.
Comments on the Quality of English Language
Language is fine
